# Determinants of Utilization of Institutional Delivery Services in Zambia: An Analytical Cross-Sectional Study

**DOI:** 10.3390/ijerph19053144

**Published:** 2022-03-07

**Authors:** Mamunur Rashid, Mohammad Rocky Khan Chowdhury, Manzur Kader, Anne-Sofie Hiswåls, Gloria Macassa

**Affiliations:** 1Department of Public Health and Sports Science, Faculty of Health and Occupational Studies, University of Gävle, Kungsbacksvägen 47, 80176 Gävle, Sweden; anne-sofie.hiswals@hig.se (A.-S.H.); gloria.macassa@hig.se (G.M.); 2Department of Epidemiology and Preventive Medicine, Faculty of Medicine, Nursing and Health Sciences, University of Monash, 553 St Kilda Road, Melbourne, VIC 3004, Australia; mohammad.chowdhury2@monash.edu; 3Department of Public Health, First Capital University of Bangladesh, Chuadanga 7200 R747, Bangladesh; 4Institute of Environmental Medicine, Karolinska Institute, Solnavägen 1, 17177 Solna Stockholm, Sweden; manzur.kader@ki.se

**Keywords:** Africa, determinants, skilled birth attendants, place of delivery

## Abstract

Institutional delivery at birth is an important indicator of improvements in maternal health, which remains one of the targets of sustainable development goals intended to reduce the maternal mortality ratio. The purpose of the present study was to identify the determinants of utilization of institutional delivery in Zambia. A population-based cross-sectional study design was used to examine 9841 women aged 15–49 years from the 2018 Zambia Demographic and Health Survey. A multiple logistic regression was applied to calculate odds ratios (ORs) with 95% confidence intervals (CIs) to identify determinants of utilization of institutional delivery. Sociodemographic factors were significantly associated with institutional delivery: woman’s (OR: 1.76; 95% CI: 1.04–2.99) and husband’s (OR: 1.83; 95% CI: 1.09–3.05) secondary/higher education, higher wealth index (OR: 2.31; 95% CI: 1.27–4.22), and rural place of residence (OR: 0.55; 95% CI: 0.30–0.98). Healthcare-related factors were also significantly associated with institutional delivery: 5–12 visits to antenatal care (OR: 2.33; 95% CI: 1.66–3.26) and measuring blood pressure (OR: 2.15; 95% CI: 1.32–2.66) during pregnancy. To improve institutional delivery and reduce maternal and newborn mortality, policymakers and public health planners should design an effective intervention program targeting these factors.

## 1. Introduction

Despite an extensive reduction in maternal mortality, maternal deaths remain high in low- and middle-income countries. Worldwide, maternal mortality dropped by 35% between 2000 and 2017, from an estimated 451,000 maternal deaths in 2000 to 295,000 in 2017 [1]. However, the highest mortality and morbidity due to pregnancy-related complications or reproductive ill-health among pregnant women have been found in Sub-Saharan Africa [2]. The maternal mortality ratio (MMR) in Sub-Saharan Africa in 2017 was very high, estimated at 542 per 100,000 live births [1]. In Zambia, the estimated MMR dropped to 213 in 2017 from 528 in 2000 [1], but maternal mortality still accounts for 10% of all deaths among women between 15 and 49 years of age [3]. In relation to Target 3.1 of the Sustainable Development Goal (SDG) 3, which is to reduce global MMR to less than 70 per 100,000 live births by 2030 [4], the estimated MMR still remains relatively high in Zambia.

Most of the maternal mortality in developing countries, including Zambia, occurs because of low levels of maternal healthcare-seeking behavior. There are pieces of evidence to suggest that inadequate antenatal care (ANC) utilization and the extremely low number of deliveries assisted by a skilled attendant are related to maternal mortality [5,6,7]. The utilization of maternal healthcare services, such as antenatal and prenatal care, and family planning have been shown to significantly reduce maternal mortality [8,9]. To ensure women’s physical and mental wellbeing, the World Health Organization [10] recommends at least eight ANC visits during pregnancy, as well as postnatal care visits at six hours, six days, six weeks, and six months.

Institutional delivery has been found to be one of the key predictors of neonatal mortality prevention, and is a recognized intervention mechanism for decreasing maternal death [11,12]. Timely institutional delivery assisted by skilled attendants can significantly reduce maternal and neonatal mortality by preventing delivery complications [13,14]. Nonetheless, a large number of women in developing countries, particularly in rural areas, lack access to institutional delivery settings, meaning that birth occurs in unsafe and unhygienic conditions [15]. Although Sub-Saharan Africa has increased its proportion of birth with skilled medical attendance, this is still below half of all birth (46.5%), compared to the observed rate of 99.5% in developed countries [16]. According to the latest 2018 Zambia Demographic and Health Survey (ZDHS 2018), the percentage of deliveries assisted by skilled birth attendants in Zambia increased from 44% in 2002 to 84% in 2018 [3]. Despite this increase, it is clear from the SDG 3 target level that more needs to be done [4]. Access to healthcare services and utilization of available skilled birth attendance can contribute to minimizing the risk of delivery complications and to preventing maternal and neonatal mortality [17].

The utilization of healthcare facilities by women and its association with institutional delivery is influenced by several factors. Previous studies worldwide have found that multilevel factors, such as personal and sociodemographic conditions, affect the decision to seek out institutional delivery with skilled birth attendants [18,19,20,21,22,23]. A recent study conducted in Southwest Ethiopia by Yoseph et al. [24] found that high wealth index, primary and above education level for the husband, age below 40 years for the woman, and the number of ANC visits were positively associated with institutional delivery. A study conducted in Eritrea demonstrated that low levels of education of women and their husbands, early initiation of ANC visits, history of giving birth in a healthcare facility, and complications during the most recent pregnancy were associated with a significant institutional delivery increase [20]. Another study found that health-facility-related factors, such as quality of care, were determinants of place of delivery [25].

This study investigated the relationship between sociodemographic and healthcare-related factors, and the utilization of institutional delivery services through the lens of Andersen’s behavioral model of health services utilization [26]. According to Andersen’s model [26], utilization of health services is determined by three sets of factors—predisposing, enabling, and need factors—that either facilitate or impede the use of healthcare services. Predisposing factors indicate existing factors before illness, which may influence an individual’s perception or beliefs of their healthcare utilization. Previous studies show that predisposing factors, such as age and gender, play a significant contribution to the use of healthcare services [24,27]. Enabling factors represent logistical aspects of obtaining healthcare. Earlier studies suggest that enabling factors, such as income and occupation, influence seeking healthcare service utilization [20,28]. Need factors refer to perceived and actual requirement for healthcare services; for instance, ANC visits and chronic disease, which may lead to a perceived need seeking healthcare services utilization [24,27]. In the present study, this conceptual framework was used as a basis for justifying the study variables and indicating potential causal pathways through which the independent variables could affect the outcome of the study (Figure 1).

Earlier studies in Sub-Saharan Africa have mainly focused on socioeconomic and demographic factors in relation to institutional delivery, and evidence-based knowledge regarding other potential determinants of healthcare utilization for childbirth in Zambia is lacking. There is a need for studies that take into account a multitude of factors, including healthcare-related factors, which could hinder or promote institutional delivery by skilled attendants. A representative population-based study, reflecting both rural and urban areas and focusing on sociodemographic and healthcare-related determinants, would help policymakers design a specific intervention approach in this regard. In order to address the knowledge gap regarding the situation in Zambia, the present study aimed to identify associations of sociodemographic and healthcare-related factors with the use of institutional delivery assisted by skilled birth attendants.

## 2. Materials and Methods

### 2.1. Study Setting

Zambia, the setting for the current study is a landlocked tropical country in Southern-Central Africa. It has ten provinces, which are divided into 117 districts. The vast majority (96%) of the population are Christian [29,30]. Per capita income was 1307 U.S. dollars in 2019, and the country achieved lower middle-income country status in 2011. The average annual gross domestic product growth rate between 2000 and 2014 was 6.8%, but economic growth declined significantly thereafter, from 4% in 2018 to 1.4% in 2019 [31,32]. In Zambia, the Ministry of Health and the Ministry of Community Development, Mother and Child Health are responsible for the delivery of healthcare services. The healthcare system is divided into three main levels: the primary level, consisting of healthcare centers and healthcare posts; the secondary level, comprising provincial/general hospitals and district hospitals; and the tertiary level, comprising tertiary teaching hospitals. Expenditure on health is almost 5% of GDP. Despite distributing 66% of the total health budget to the subnational level (provincial and district), Zambia has still socioeconomic inequalities in public health service utilization [33,34]. The private healthcare sectors remain a significant provider of skilled birth attendance (i.e., 98.1% in private healthcare sector, whereas 95% in public healthcare). This sector charges high delivery fees, whereas child-birth at public hospitals is almost free of charge [3,35]. Health cost inequalities are also seen between rural and urban areas, due to differing distances to the nearest health facility.

### 2.2. Data Source

This study used secondary cross-sectional data extracted from the 2018 Zambia Demographic and Health Survey (ZDHS) [3], which was implemented by the Zambia Statistics Agency (ZamStats, Lusaka, Zambia) in collaboration with the Ministry of Health. The survey is nationally representative and was conducted from 18 July 2018 to 24 January 2019 under the auspices of the United States Agency for International Development together with the technical assistance of Inner City Fund (ICF) International, USA. Four questionnaires were used in the 2018 ZDHS: a household questionnaire, a woman’s questionnaire, a man’s questionnaire, and a biomarker questionnaire [3]. The woman’s questionnaire was used to collect information from all eligible women aged 15–49 years via interviews conducted by trained field workers. 

Women were asked questions on the following topics: socioeconomic and demographic factors (e.g., age, education, occupation, place of residence, religion), healthcare-related matters such as breastfeeding, maternal care and child care (antenatal, delivery, and postnatal), reproductive history, family planning methods, immunizations and illnesses, fertility preferences, and awareness of AIDS, other sexually transmitted infections, domestic violence, and women’s empowerment [3].

### 2.3. Study Design and Sample

A community-based cross-sectional study was conducted to assess determinants of the preference for institutional delivery based on the women’s entire birth file in the 2018 ZDHS data.

All women aged 15–49 who were either permanent residents in the selected household or visitors who stayed in the household on the night before the survey were eligible to be interviewed for the 2018 ZDHS. A two-stage stratified sampling method was employed, covering the demographics of women across the country. The first stage of the survey involved selecting a total of 545 sample points (clusters) consisting of enumeration areas. These enumeration areas were chosen with a probability proportional to their population size within each sampling stratum. In the second stage, systematic sampling was used to obtain an average of 133 households from each cluster. A fixed number of 25 households were then selected from each cluster through an equal probability systematic selection process; thus, the 2018 ZDHS covered 13,625 households in total. The sample size was representative at the national, urban, rural, and provincial level.

Of the initially selected 13,625 households, 12,831 households were successfully interviewed, from these a total of 14,189 eligible women aged 15–49 years with a child aged 0–5 years were identified for an individual interview. Finally, out of the 14,189 women, 13,683 women were interviewed, yielding a response rate of 96%. A similar response rate was observed in rural and urban areas. The women who were interviewed had experienced 38,446 births in total. For the present study, the last birth within the past five years was considered, resulting in 9841 deliveries within this period being included in the analyses.

### 2.4. Variable Measurement

#### 2.4.1. Outcome Variable

The outcome variable of institutional delivery service utilization was dichotomized as yes or no. Giving birth in any healthcare facility (e.g., government hospital, private hospital, or healthcare and family welfare center) while assisted by skilled birth attendants (a delivery with the assistance of a skilled health professional, e.g., mid-wife or trained obstetric doctor/nurse) was categorized as an institutional delivery, and all other births were categorized as non-institutional deliveries (i.e., occurring at home by non-professional facilities).

#### 2.4.2. Explanatory Variables

The following explanatory variables were selected based on previous studies [19,20,21,36,37] and the availability of relevant data, and categorized based on previous evidence and distribution of sample size in each category.

Sociodemographic variables were categorized as follows: age into four groups (15–24, 25–30, 31–39, 40–49); place of residence into urban or rural; woman’s education into no formal education, primary, or secondary/higher; husband’s education into no education, primary, or secondary/higher; religion into catholic, protestant, or other; woman’s current working status into yes or no; and the number of children into 1, 2, 3 or more. The variable wealth index was categorized into poor, middle, or high by scoring the consumer goods owned by the household, ranging from a television to a bicycle or car, and housing characteristics such as the source of drinking water, toilet facilities, and flooring materials. Principal components analysis was used to assign the indicator weights in order to create these scores [3,38]. Contraception use decision was into woman’s decision, husband’s decision, or joint decision.

Healthcare-related variables were coded as follows: number of ANC visits into 1–3, 4, or 5–12; healthcare facility into yes or no, determined by whether the woman had access to a healthcare facility in the past 12 months. Blood pressure into yes or no, determined by having had regular blood pressure measurement during pregnancy; anemia into yes or no, determined by whether the woman had given a blood sample for an anemia test.

### 2.5. Statistical Analyses

Because the ZDHS data were not randomly collected, the dataset was weighted using the sample weight variable provided in the dataset to normalize the data at the national level and to minimize the standard errors. Complex sampling design and sampling weight were considered for all analyses. In order to create the complex sampling design, a two-step process was followed; first, the complex samples were created as a plan file, and then all the analyses were run using the plan file through a complex sample package to account for the sample design. Three variables were needed to create the complex sample package: (i) primary sampling units, with probability proportional to size within each stratum and forming sample clusters based on the 2010 Zambian census frame; (ii) sample strata, comprising sample clusters that had been stratified into rural and urban before drawing a representative sample; and (iii) sample weights. Sample weights were calculated based on sampling probability for each sampling stage separately and for each cluster by adjusting for non-response; the sampling weights are then normalized to obtain the final standard weights.

A descriptive analysis was used to present the basic characteristics of the sample. Pearson’s chi-squared test was used to check for variables associated with the choice of institutional delivery, and variables with a *p*-value ≤ 0.5 were then entered into regression models. Both univariate and multivariate logistic regression analyses were conducted to identify the associations between the explanatory variables and institutional delivery. Odds ratios (ORs) with their corresponding 95% confidence intervals (CIs) were calculated for both univariate and multivariate logistic regression analyses. The level of significance for all analyses was set at *p* ≤ 0.05.

Variance inflation factors (VIFs) were estimated in order to check collinearity between explanatory variables, with a VIF value greater than 10 assumed to indicate high collinearity. The models checked for goodness-of-fit using the Nagelkerke pseudo *R*^2^. Data analyses were performed using version 27 of IBM SPSS Statistics for Windows (SPSS Inc., Chicago, IL, USA).

### 2.6. Ethics Approval and Consent to Participants

The analyses were performed using publicly available data from the demographic health surveys, for which all participants had given their informed consent. In order to conduct the present study, ethical approval for using the data was obtained from ICF International Rockville, Maryland, USA, in October 2020. All methods in this study were carried out in accordance with relevant guidelines and regulations.

## 3. Results

Table 1 represents background characteristics of the study population. Among the 9841 women, 84.9% had an institutional delivery attended by skilled healthcare professionals.

Approximately three-quarters of the women were between 15 and 34 years of age, and more than 60% were of rural origin. The education rate was higher among the husbands (93.3%) than among the women (89.9%). Regarding the wealth index, 47.1% of the women were classified as economically poor, 18.5% as middling, and 34.4% as rich. More than 60% of women had attended 4–12 ANC visits during pregnancy, while 35.2% had attended only 1–3. Most of the women were Christian (96%), were not currently working (52.8%), had two children (44.7%), had access to a healthcare facility (73.3%), had undergone blood pressure measurement (95.2%), and had been tested for anemia (28.2%) during pregnancy (Table 1). The numbers of deliveries at home and at an institution, divided by place of residence (urban vs. rural) are presented in Figure 2.

The comparison of women with and without institutional delivery revealed that the woman’s age, place of residence, parental education, number of children, wealth index, contraception use decision, ANC visits, healthcare facility, and blood pressure differed significantly between groups. Conversely, religion, working status, and anemia did not show significant differences between groups (Table 2). The value of VIF between the independent variables was less than 1.5, indicating that no specific variables were highly collinear in the prediction models [39].

The results of the univariate and multivariate regression analyses are summarized in Table 3. Of the seven sociodemographic and four healthcare-related variables that were set to be assessed, all except two categories—four ANC visits and husband’s decision on contraception use—and anemia were found to be significant determinants of institutional delivery using univariate analysis.

In the multivariate regression analyses, the results showed that women from rural areas were less likely to have delivered at an institution with skilled attendants in comparison to those living in urban areas (OR: 0.55; 95% CI: 0.30–0.98). Women with secondary/higher education (OR: 1.76; 95% CI: 1.04–2.99) and belonging to the higher economic stratum (OR: 2.31; 95% CI: 1.27–4.22) were more likely to have delivered at an institution with skilled attendants, compared to those who had no formal education and belonged to the poor economic stratum. Similarly, women whose husbands had a secondary or higher education had 83% higher utilization of institutional delivery compared to women whose husbands had no formal education (OR: 1.83; 95% CI: 1.09–3.05). Women who received 5–12 ANC visits were 2.33 times more likely to have delivered at an institution with skilled attendants (OR 2.33; 95% CI: 1.66–3.26). Finally, women who had their blood pressure measured (OR: 2.15; 95% CI: 1.32–2.66) during pregnancy had an increased likelihood of delivering at institution with skilled attendants compared to those who did not have their blood pressure measured (Table 3).

## 4. Discussion

The present study revealed that sociodemographic factors—women’s and their husbands’ secondary/higher level of education and women’s higher wealth index—were positively associated with institutional delivery; conversely, living in rural areas was negatively associated with institutional delivery. The study also revealed an association with healthcare-related factors, in that woman who attended at least five ANC visits and who had measured their blood pressure during pregnancy, were more likely to deliver at a healthcare facility.

Women living in rural areas were less likely to give birth in a healthcare facility compared to women living in urban areas. This finding is similar to results from previous studies [22,37,40] and other studies conducted in Sub-Saharan Africa [7,41], including a systematic review and meta-analysis [42]. One possible explanation is that women living in rural areas have less access to antenatal obstetric and postnatal care services and poor access to transportation, whereas women living in urban areas have greater access to maternal healthcare services near their home, as well as to transportation [43]. Consequently, women in rural areas do not utilize family planning, antenatal and other services as much as women in urban areas do [34,35]. Moreover, women who live in urban areas are closer to information about the health benefits of institutional delivery [35]; conversely, rural women are sometimes affected by cultural taboos concerning the choice of place of delivery. Because inequality regarding the choice of place of delivery exists between rural and urban areas, further study is warranted by stratifying the analysis between urban and rural women in this regard.

The education level was found to be an important determinant of institutional delivery. Women with a secondary/higher education were more likely to have delivered at a healthcare facility compared to women with no formal education or primary education. Similarly, having a husband with a secondary/higher education also played an important role in determining the place of delivery. This finding is supported by previous studies showing that the education level of the woman and husband had a significant effect on the choice of place of delivery [19,20,22,25,44]. The reason for this could be that education improves access to information and health education, knowledge of services, and control over resources; if so, then this finding implies a change in women’s attitudes toward a preference for delivery at a healthcare facility with skilled attendants [7,36,45].

In line with previous studies [19,24,46], the present study found that women with a higher wealth index were more likely than their less-privileged counterparts to have delivered with the assistance of skilled birth attendants. This may be due to the costs, in that having a higher wealth index may help women cover all the expenses required for delivery at an institution with skilled attendants. Moreover, better economic status may increase health-seeking behavior and autonomy of healthcare decision-making, which in turn may have a positive influence on overall healthcare utilization [24]. This finding, therefore, reinforces the notion that economic status affects the choice to deliver at a healthcare facility.

The odds of institutional delivery service utilization were higher among women who had visited ANC five or more times than among women who had fewer visits. This finding is consistent with previous studies conducted in other African countries [20,36,40], including a systematic review and meta-analysis [47]. A possible explanation may be that having more ANC visits works as a platform for emphasizing the importance of institutional delivery and increases women’s awareness of the risks of pregnancy and childbirth. Moreover, women attending more ANC visits are more likely to be informed about complications associated with home delivery, which motivates them to deliver at an institution with skilled attendants. The present findings suggest that the new WHO guidelines, which recommend at least eight ANC visits for every pregnant woman [10], will contribute to increasing the rate of healthcare facility delivery.

Finally, the present study showed that women whose blood pressure was measured during pregnancy had an increased likelihood of delivering at an institution with skilled attendants in comparison with women who did not have their blood pressure measured. One possible explanation for this finding is that women who had their blood pressure measured may have experienced pregnancy-related complications [48], which in turn made them vigilant and encouraged them to deliver their baby at an institution with skilled attendants. An earlier study looking at high blood pressure as a complication of pregnancy found that the presence of such complications was significantly associated with healthcare facility delivery [37]. Further research is recommended in this regard.

The present study was based on a large sample. It had a high response rate (96%) and a wide range of variables, from both sociodemographic and healthcare perspectives. The data were analyzed by considering cluster effects using survey weights that represented the whole country, both urban and rural areas, to ensure the precision of the estimates. This increases the external validity of the findings. Because the study analyzed data from a national survey, the findings are applicable to the whole country of Zambia and can be generalized to other low- and middle-income countries with similar socio-demographic characteristics and healthcare setting. However, the secondary nature of the data and the cross-sectional design of the study mean that the associations do not indicate causation. Moreover, there was a possibility of recall bias, in that some of the information collected was based on events from the past.

## 5. Conclusions

The analyses of the 2018 ZDHS showed that sociodemographic factors (e.g., place of residence, parental secondary/higher education, and higher wealth index) and healthcare-related factors (e.g., 5–12 ANC visits, and blood pressure measurement during pregnancy) were significantly associated with institutional delivery. Therefore, policymakers and public health planners should design an effective intervention program to scale up maternal health programs, including institutional delivery, targeting disadvantaged groups (poor, rural residents, and low or no education), and encouraging increased use of ANC services. Implementing an initiative to measure blood pressure measurement during pregnancy would also be useful for improving institutional delivery and reducing maternal and newborn mortality in the study areas. In addition, highlighted factors from this study could be considered by the Ministry of Health in Sub-Saharan Africa, particularly in Zambia, for developing an effective intervention program. Because this study investigated factors in general—i.e., data from both urban and rural, further study is warranted to examine factors by stratifying analysis between urban and rural.

## Figures and Tables

**Figure 1 ijerph-19-03144-f001:**
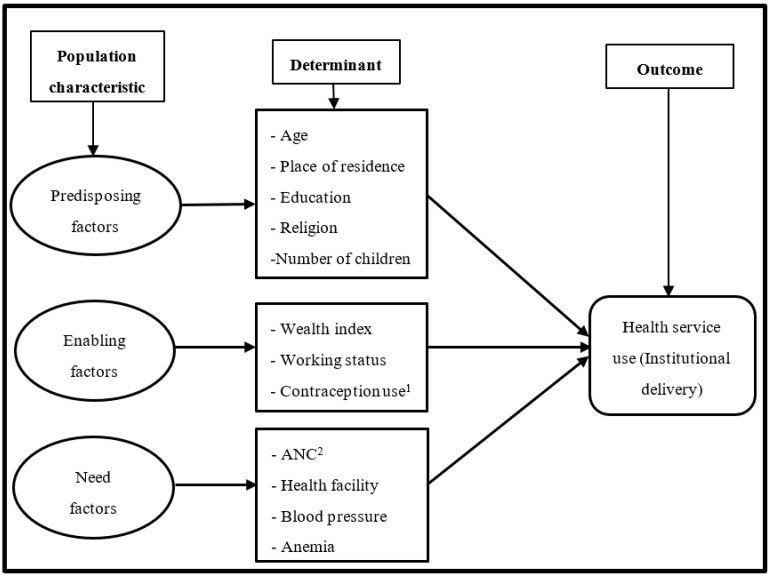
The conceptual framework for health services utilization is modified from Andersen’s behavioral model. ^1^ Contraception use decision; ^2^ number of antenatal care visits during pregnancy.

**Figure 2 ijerph-19-03144-f002:**
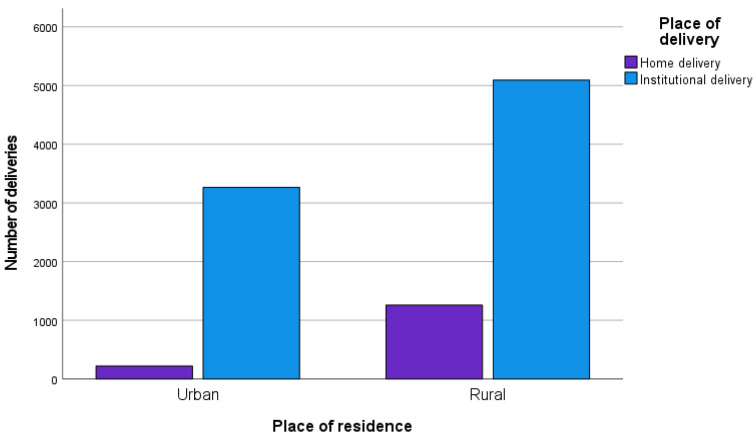
Delivery pattern by place of residence.

**Table 1 ijerph-19-03144-t001:** Background characteristics of the study population (*n* = 9841).

Variable	Category	Frequency	Percentage
Woman’s age			
	15–24	3405	34.6
	25–34	4224	42.9
	35–39	1421	14.4
	40–49	791	8.0
Place of residence			
	Urban	3489	35.5
	Rural	6352	64.5
Woman’s education			
	No formal education	996	10.1
	Primary	5008	50.9
	Secondary/higher	3837	39.0
Husband’s education			
	No education	491	6.7
	Primary	2909	39.4
	Secondary/higher	3978	53.9
Religion			
	Catholic	1569	15.9
	Protestant	8111	82.4
	Other	161	1.6
Working status ^1^			
	No	5193	52.8
	Yes	4648	47.2
Number of children ^2^			
	1	3672	38.9
	2	4209	44.7
	3 or more	1545	16.4
Wealth index			
	Poor	4634	47.1
	Middle	1823	18.5
	Rich	3385	34.4
Contraception use decision			
	Woman’s decision	609	14.8
	Husband’s decision	475	11.5
	Joint decision	3038	73.7
ANC ^3^			
	1–3 visits	2531	35.2
	4 visits	2358	32.8
	5–12 visits	2293	31.9
Healthcare facility			
	No	2632	26.7
	Yes	7209	73.3
Blood pressure ^4^			
	No	348	4.8
	Yes	6897	95.2
Anemia			
	No	6875	71.8
	Yes	2706	28.2
Institutional delivery ^5^			
	No	1482	15.1
	Yes	8359	84.9

^1^ Woman’s current working status: yes = working, no = not working; ^2^ number of children ever born; ^3^ ANC = number of antenatal care visits during pregnancy. ^4^ Whether the woman had regular blood pressure measurement during pregnancy. ^5^ Institutional delivery: yes = women with institutional delivery, no = women without institutional delivery.

**Table 2 ijerph-19-03144-t002:** Characteristics of women with and without institutional delivery across the explanatory variables (*n* = 9841).

Variable	Institutional Delivery (ID)				*p*-Value
	Women with ID		Women without ID		
	%	(95% CI)	%	(95% CI)	
Sociodemographic variables					
Woman’s age					
15–24	36.0	(34.5–37.6)	26.7	(24.1–29.6)	<0.001
25–34	42.5	(40.9–44.1)	45.2	(42.0–48.4)	
35–39	14.1	(12.7–15.6)	16.3	(14.2–18.7)	
40–49	7.4	(6.6–8.2)	11.7	(9.8–14.1)	
Place of residence					
Urban	39.1	(35.5–42.8)	15.0	(11.3–19.7)	<0.001
Rural	60.9	(57.2–64.5)	85.0	(80.3–88.7)	
Woman’s education					
No formal education	8.2	(7.2–9.2)	21.3	(17.1–26.2)	<0.001
Primary	49.1	(47.1–51.1)	61.1	(56.9–65.1)	
Secondary/higher	42.8	(40.8–44.8)	17.6	(14.5–21.3)	
Husband’s education					
No education	5.5	(4.7–6.5)	12.6	(9.4– 16.7	<0.001
Primary	36.6	(34.7–38.6)	54.8	(50.3–59.2)	
Secondary/higher	57.9	(55.8–60.0)	32	(28.3–37.2)	
Religion					
Catholic	16.1	(14.4–17.9)	15.2	(12.1–19.0)	0.87
Protestant	82.3	(80.3–48.1)	83.2	(79.5–86.4)	
Other	1.7	(1.1–2.4)	1.5	(0.8–3.0)	
Working status ^1^					
No	52.6	(50.6–54.7)	53.5	(49.0–57.9)	0.71
Yes	47.4	(45.3–49.4)	46.5	(42.1–51.0)	
Number of children ^2^					
1	40.9	(39.2–42.7)	28.2	(25.4–31.3)	<0.001
2	44.0	(42.4–45.6)	48.3	(45.1–51.6)	
3 or more	15.1	(13.6–16.8)	23.4	(20.3–26.8)	
Wealth index					
Poor	43.1	(40.1–46.1)	69.7	(65.2–73.8)	< 0.001
Middle	18.8	(17.2–20.5)	16.9	(14.4–19.8)	
Rich	38.1	(35.3–41.0)	13.4	(10.3–17.2)	
Contraception use decision					
Woman’s decision	14.6	(12.8–16.6)	16.3	(12.6–20.7)	0.05
Husband’s decision	11.0	(9.5–12.8)	14.7	(11.2–19.2)	
Joint decision	74.4	(72.0–76.7)	69.0	(63.9–73.7)	
Healthcare-related variables					
ANC ^3^					
1–3 visits	33.3	(31.7–35.1)	48.7	(44.5–53.0)	<0.001
4 visits	33.8	(32.3–35.2)	26.2	(23.3–29.3)	
5–12 visits	32.9	(31.3–34.6)	25.0	(21.5–29.0)	
Healthcare facility					
No	25.8	(23.8–27.9)	32.0	(27.6–36.7)	0.002
Yes	74.2	(72.1–76.2)	68.0	(63.3–72.4)	
Blood pressure ^4^					
No	3.8	(3.2–4.6)	11.8	(8.2–16.8)	<0.001
Yes	96.2	(95.4–96.8)	88.2	(83.2–91.8)	
Anemia					
No	71.9	(70.3–73.5)	70.9	(67.5–74.1)	0.54
Yes	28.1	(26.5–29.7)	29.1	(25.9–32.5)	

^1^ Woman’s current working status: yes = working, no = not working); ^2^ number of children ever born; ^3^ ANC = number of antenatal care visits during pregnancy. ^4^ Whether the woman had regular blood pressure measurement during pregnancy; CI = confidence interval.

**Table 3 ijerph-19-03144-t003:** Logistic regression showing the crude and adjusted odds ratio (OR) with 95% confidence intervals (CIs) for utilization of institutional delivery services among women in Zambia.

Variable	Crude Analysis		Adjusted Analysis	
	OR	95% CI	OR	95% CI
Sociodemographic variables				
Woman’s age				
15–24	1		1	
25–34	0.70	0.61–0.80	1.11	0.66–1.87
35–39	0.64	0.54–0.76	1.05	0.69–1.62
40–49	0.48	0.38–0.57	0.79	0.49–1.28
Place of residence				
Urban	1		1	
Rural	0.28	0.24–0.32	0.55	0.30–0.98
Women’s education				
No education	1		1	
Primary	2.10	1.80–2.44	1.38	0.97–1.99
Secondary/higher	6.32	5.27–7.60	1.76	1.04–2.99
Husband’s education				
No formal education	1		1	
Primary	2.66	2.17–3.28	1.29	0.93–1.78
Secondary/higher	4.05	2.77–5.91	1.83	1.09–3.05
Number of children ^1^				
1	1		1	
2	0.71	0.58–0.87	0.83	0.56–1.22
3 or more	0.45	0.36–0.56	0.78	0.50–1.19
Wealth index				
Poor	1		1	
Middle	2.56	1.92–3.41	1.75	0.96–3.19
Rich	4.60	3.38–6.26	2.31	1.27–4.22
Contraception use decision				
Woman’s decision	1		1	
Husband’s decision	1.20	0.90–1.61	1.11	0.83–1.67
Joint decision	1.44	1.04–1.99	1.23	0.75–1.76
Healthcare-related variables				
ANC ^2^				
1–3 visits	1		1	
4 visits	1.02	0.82–1.27	1.20	0.85–1.71
5–12 visits	1.92	1.53–2.41	2.33	1.66–3.26
Healthcare facility				
No	1		1	
Yes	1.35	1.12–1.64	1.21	0.84–1.76
Blood pressure ^3^				
No	1		1	
Yes	3.37	2.24–5.08	2.15	1.32–2.66
Anemia				
No	1		1	
Yes	0.95	0.84–1.08	1.15	0.84–1.56

^1^ Number of children ever born; ^2^ ANC= number of antenatal care visits during pregnancy. ^3^ Whether the woman had regular blood pressure measurement during pregnancy; 1 = reference group.

## Data Availability

The datasets used for this study are publicly available from the DHS Program website http://dhsprogram.com/data (accessed on 25 February 2021).

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
