# Peer review of "Determinants of Utilization of Institutional Delivery Services in Zambia: An Analytical Cross-Sectional Study"

_ijerph, 2022, doi:10.3390/ijerph19053144_

Round 1

Reviewer 1 Report

This is an excellent analysis of use of institutional delivery services among childbearing women in Zambia.

It addresses a serious issue in global maternal health, access and use of delivery services involving skilled medical personnel in an institutional healthcare setting.

It uses high quality, large sample data that is appropriate to the task.

It uses appropriate statistical methods, with clear theoretical and methodological justification for the choice and coding of variables. These are well justified from the prior literature, and descriptive analysis.

The results are important for understand which women access institutional delivery services, with clear policy implications for Zambia specifically, but for Sub-Subsaharan African and lower-income countries generally as well.

Limitations are discussed.

In the discussion of Table 1, at p. 7 @243, the authors refer to "literacy rate," but the table discusses levels of education -- the text and table should use the same terminology.

Thanks for the opportunity to review this paper.

Author Response

Thank you for the feedback and comments. We highly appreciate it. 

Comment:

In the discussion of Table 1, at p. 7 @243, the authors refer to "literacy rate," but the table discusses levels of education -- the text and table should use the same terminology.

Author's reply:

Thank you. The text has now been changed according to the Table.

Reviewer 2 Report

This is a very well designed, executed study with large implications for maternal health policy in Zambia. The findings, while not particularly surprising (at least to this reviewer), are important for the Ministry of Health to acknowledge and use as a basis for policy changes to improve the odds that women in Zambia (urban and rural) will deliver in an institution. That being said, the trend in many countries has been for women with uncomplicated pregnancies to deliver at home with a trained midwife in attendance. This should be acknowledged in the paper--not every woman has to or should deliver in hospital. 

On a positive note, the response rate is hugely impressive! This reviewer is amazed that 13, 683 women were interviewed. Goodness, how long did that take???

It would be helpful to the reader if the authors defined what they mean by 'skilled birth attendants'. It also would be helpful if the authors give an estimate of the number of Obstetricians and midwives in the country, and, if possible, stratified by rural and urban. Are there 'sufficient numbers' of trained birth attendants in the country? in rural areas?

The authors mention that "the private healthcare sector remains a significant provider of skilled birth attendance". What proportion of deliveries are private vs public? Please stratify by rural/urban if you can.

While the findings are not necessarily surprising (ie, more rural women than urban women do not delivery in an institution; women with lower wealth index do not deliver in an institution; etc), the findings from this study are important for planning purposes. The authors should present suggestions for the MoH to consider, based on the data presented. Be mindful, however, that having a skilled, trained midwife deliver the baby in the woman's home is not necessarily "bad:. All deliveries need not take place in an institution.

Overall, this study was well designed and the findings presented clearly. 

Author Response

Thank you for the feedback and comments. We highly appreciate it. 

Comment 1:

This is a very well designed, executed study with
large implications for maternal health policy in
Zambia. The findings, while not particularly
surprising (at least to this reviewer), are important
for the Ministry of Health to acknowledge and use
as a basis for policy changes to improve the odds
that women in Zambia (urban and rural) will
deliver in an institution. That being said, the trend
in many countries has been for women with
uncomplicated pregnancies to deliver at home
with a trained midwife in attendance. This should
be acknowledged in the paper--not every woman
has to or should deliver in hospital. 

Author's reply:

Thank you for the thoughtful insights, and we
agree with the reviewer. However, in ZDHS
data, women who gave birth with the
assistance of skilled attendance, e.g., a trained
midwife, were comprised of institutional
delivery service utilization, which has now
been clarified in the Methods section (page 4–
5; lines 174–178).

Comment 2:

It would be helpful to the reader if the authors
defined what they mean by 'skilled birth
attendants'. It also would be helpful if the authors
give an estimate of the number of Obstetricians
and midwives in the country, and, if possible,
stratified by rural and urban. Are there 'sufficient
numbers' of trained birth attendants in the
country? in rural areas? 

Author's reply:

The skilled birth attendant is defined as a
delivery that occurs with the assistance of a
skilled health professional, e.g., mid-wife or
trained obstetric doctor/nurse. We have now
added texts on that in page 4–5; lines 177–178.
Unfortunately, we did not find an exact
estimation of the number of Obstetricians and
midwives in Zambia. However, according to
the world bank data, a total of 1.02 nurses and
midwives (per 1000 people) (2018), and 0.09
physicians (per 1000 people) (2016). This
means trained attendants may not be sufficient
in both rural and urban

Comment 3

The authors mention that "the private healthcare
sector remains a significant provider of skilled
birth attendance". What proportion of deliveries
are private vs public? Please stratify by
rural/urban if you can.

Author's reply:

The proportion of deliveries in public and
private health facilities was 95.0% and 98.1%,
respectively, which has now been added in the
Methods section (page 3–4; lines 27–28)
Unfortunately, information between urban and
rural in this regard was not available. 

Comment 4: 

While the findings are not necessarily surprising
(ie, more rural women than urban women do not
delivery in an institution; women with lower
wealth index do not deliver in an institution; etc),
the findings from this study are important for
planning purposes. The authors should present
suggestions for the MoH to consider, based on the
data presented. Be mindful, however, that having
a skilled, trained midwife deliver the baby in the
woman's home is not necessarily "bad:. All
deliveries need not take place in an institution.

Author's reply:

We agree with the reviewer. Now the text has
been added for taking the findings into account
by the Ministry of Health in Sub-Saharan
Africa, particularly Zambia (page 11; lines
375– 377).

Reviewer 3 Report

The purpose of the present study was to identify determinants of utilization of institutional delivery in Zambia. 
In the paper, the research method and the entire research procedure was correctly designed and conducted. However, I suggest that at least one research hypothesis should be added in Material and method and Introduction and then confirmed or falsified. 
The analyses presented can be helpful to health care policy makers in Africa. I suggest in the conclusion to add information about the limitations of the present study and guidelines for further research. 

Author Response

Thank you for the feedback and comments. We highly appreciate it.

Comment 1: 

In the paper, the research method and the entire
research procedure was correctly designed and
conducted. However, I suggest that at least one
research hypothesis should be added in Material
and method and Introduction and then confirmed
or falsified.

Author's reply:

We highly appreciate the reviewer’s comment.
However, this is an observational study to
investigate an association between several
variables and an outcome and that is clear from
the aim of the study and methods description;
therefore, we decide not to add a hypothesis
for this study. 

Comment 2:

The analyses presented can be helpful to health
care policy makers in Africa. I suggest in the
conclusion to add information about the
limitations of the present study and guidelines for
further research.

Author's reply:

We agree with the reviewer. The text has been
added in this regard (page 11; lines 377– 379).